# Extension of the Rigid-Constraint Method for the Heuristic Suboptimal Parameter Tuning to Ten Sensor Fusion Algorithms Using Inertial and Magnetic Sensing

**DOI:** 10.3390/s21186307

**Published:** 2021-09-21

**Authors:** Marco Caruso, Angelo Maria Sabatini, Marco Knaflitz, Ugo Della Croce, Andrea Cereatti

**Affiliations:** 1PolitoBIOMed Lab—Biomedical Engineering Lab, Politecnico di Torino, 10129 Torino, Italy; marco.knaflitz@polito.it; 2Department of Electronics and Telecommunications, Politecnico di Torino, 10129 Torino, Italy; andrea.cereatti@polito.it; 3Department of Excellence in Robotics & AI, The BioRobotics Institute, Scuola Superiore Sant’Anna, 56127 Pisa, Italy; angelo.sabatini@santannapisa.it; 4Department of Biomedical Sciences, University of Sassari, 07100 Sassari, Italy; dellacro@uniss.it

**Keywords:** orientation estimation, sensor fusion, MIMU, filter parameter tuning, kalman filter, complementary filter, optimal parameter, suboptimal parameter, wearable sensors, human motion analysis, MARG, AHSR

## Abstract

The orientation of a magneto-inertial measurement unit can be estimated using a sensor fusion algorithm (SFA). However, orientation accuracy is greatly affected by the choice of the SFA parameter values which represents one of the most critical steps. A commonly adopted approach is to fine-tune parameter values to minimize the difference between estimated and true orientation. However, this can only be implemented within the laboratory setting by requiring the use of a concurrent gold-standard technology. To overcome this limitation, a Rigid-Constraint Method (RCM) was proposed to estimate suboptimal parameter values without relying on any orientation reference. The RCM method effectiveness was successfully tested on a single-parameter SFA, with an average error increase with respect to the optimal of 1.5 deg. In this work, the applicability of the RCM was evaluated on 10 popular SFAs with multiple parameters under different experimental scenarios. The average residual between the optimal and suboptimal errors amounted to 0.6 deg with a maximum of 3.7 deg. These encouraging results suggest the possibility to properly tune a generic SFA on different scenarios without using any reference. The synchronized dataset also including the optical data and the SFA codes are available online.

## 1. Introduction

The estimation of the orientation in space of a body using a sensor fusion algorithm (SFA) applied to the recordings of a wearable Magnetic and Inertial Measurement Unit (MIMU) requires the proper setting of the values of its parameters. [1,2,3,4,5,6,7]. A common method to set the parameter values consists in exploiting the ground-truth information to minimize the overall difference between the estimated and the true orientation for a given recording (optimal working condition).

However, the latter approach may be unfeasible since gold standard such as stereophotogrammetric system (SP) are rarely available out of specialized human movement laboratories. Recent works have confirmed the crucial role played by the criteria used to select appropriate parameter values [8,9,10,11]. It has also been shown that several intrinsic and extrinsic factors should be considered when performing this selection. Among them, the most critical ones are sensor noise characteristics, amplitude of motion, intensity of ferromagnetic disturbances and time required for the algorithm to converge. In light of these considerations, alternative strategies for selecting parameter values without orientation references are needed. To address this problem, we have recently presented in [9] a rigid-constraint method (RCM) for a sub-optimal estimation of the values of the single parameter (β) of the sensor fusion algorithm by Madgwick et al., [12] using a heuristic procedure which does not rely on an orientation reference. The assumption on which the method relies, is that two MIMUs aligned on a rigid body must have a null orientation difference during the movement. The orientation of the two MIMUs was computed for several values of β, separately. The main finding was that the value of β which minimizes the relative orientation difference between the MIMUs also guarantees small absolute orientation errors: the difference between the obtained and the optimal error (i.e., minimum) was 1.5 deg on average and 2.5 deg at most. This can be justified by considering that the error and disturbance characteristics affecting the sensors embedded in the two MIMUs are, in general, independent, as discussed more in detail in Section 2.1 “RCM description”. To the best of our knowledge, the RCM is the only method in the literature that specifically addressed the problem of estimating the most suitable parameter values (suboptimal) without relying on the use of a gold standard.

In this framework, this paper aims at verifying the generalizability of the RCM presented in [9] to different sensor fusion algorithms including five complementary filters (CFs) and five Kalman filters (KFs). To this end, nine experimental scenarios were considered including six MIMUs from three different manufacturers and three rotation rate magnitudes. The motions consisted in a mix of 2D and 3D rotations. For each scenario, the absolute errors corresponding to the selected suboptimal values were compared with those obtained under optimal working conditions. When this difference was lower or equal than 0.5 deg, the RCM was considered completely successful in estimating the suboptimal parameter values corresponding to the optimal orientation error for that specific SFA and experimental scenario.

## 2. Materials and Methods

### 2.1. Optimal vs Suboptimal Working Conditions

Optimal working conditions can be considered the “best case scenario” in which the parameter values of each SFA are set to provide the lowest absolute orientation error (i.e., the best performance achievable) with respect to the orientation reference provided by a gold standard system [10]. However, this can be achieved only when the experiments are conducted inside the laboratory, thus precluding the use of MIMUs for an unconstrained monitoring of patients in their free-living environment, where the orientation reference is not available.

On the other hand, suboptimal working conditions refer to the case in which suitable parameter values for each experimental scenario are selected without using any orientation reference. Under reasonable assumptions, ad-hoc methods are developed with the purpose of obtaining errors close to those obtained using optimal values. In this favorable but realistic situation, the orientation errors of each SFA are in general not coincident with the minimum achievable, as expected. In this work, the RCM is used for the selection of the suboptimal values of each of the 10 SFAs.

#### RCM Description

The target of the RCM proposed in [9] is the estimation of the most suitable parameter values (i.e., suboptimal) for a given filter and experimental scenario without relying on any orientation reference. The RCM assumption is that the combination of parameter values that provide the minimum relative orientation difference also provide an acceptable absolute orientation error (see the Glossary for the rigorous definitions of absolute orientation error and relative orientation difference). The hypothesis under analysis holds when the source of errors affecting gyroscope, accelerometer and magnetometer are different between the MIMUs. In fact, the biases affecting two different gyroscopes are uncorrelated, being them mounted on different chips, with different sensing elements and different read-out and conditioning circuits [13]. In addition, when the rotation component may not be neglected and ferromagnetic disturbances are present, for accelerometer and magnetometer, difference between the corresponding signals of the MIMUs are expected to grow as the relative distance increases (see Figures 2 and 3. of [9]). This work aims at demonstrating that the same method may be generalized to estimate the suboptimal parameter values for a generic SFA with a given number of parameters.

### 2.2. Selected SFAs

The same SFAs implemented in [10] are considered to test the validity of the RCM. The MATLAB (R2020a, The MathWorks Inc., Natick, MA, USA) implementations for each SFA were made available on GitHub (at https://github.com/marcocaruso/sensor_fusion_algorithm_codes, accessed on 21 June 2021) and also on MATLAB Exchange (at https://it.mathworks.com/matlabcentral/fileexchange/90351-orientation_estimation_sensor_fusion_algorithm_codes, accessed on 4 May 2021). In summary, five complementary filters and five Kalman filters are considered, chosen among the most popular:

Complementary filters:Mahony et al., 2008 [14] (MAH), with 2 parameters;Madgwick et al., 2011 [12] (MAD), with 1 parameter;Valenti et al., 2015 [15] (VAC), with 9 parameters;Seel et al., 2017 [16] (SEL), with 4 parameters;MATLAB complementary filter R2020a (MCF), the implementation of VAC by the MathWorks with only two parameters.

Kalman filters:Sabatini 2011 [17] (SAB), with 6 parameters;Ligorio and Sabatini 2015 [18] (LIG), with 6 parameters;Valenti et al., 2016 [19] (VAK), with 3 parameters;Guo et al., 2017 [20] (GUO), with 3 parameters;MATLAB Kalman filter R2020a (MKF), the implementation by MathWorks of the filter by Roetenberg et al., 2005 [21], with 8 parameters.

For each SFA the two most influencing parameters (selected from those that caused the greatest variation in orientation as their values changed) were tuned following a grid-search approach. The choice of limiting the tuning to at most two parameters was a compromise between the dimension of the search-space and the computational time. The parameter related to the weight given to the gyroscope was always tuned, when exposed by the SFA, since the angular velocity is the most important information in a sensor fusion framework [17]. The additional parameter tuned, when relevant, was selected to weight differently the accelerometer or magnetometer readings, depending on the SFA. As a general consideration, it is not advisable to set the accelerometer and magnetometer related parameters based on the sensor electrical noise only. In fact, the latter is usually negligible compared to errors due to linear accelerations and the additional magnetic fields. The remaining parameters have been set to the default values reported in the original articles.

Table 1 reports the details of the parameters tuned for each SFA together with the default values provided by the SFA authors.

### 2.3. Experimental Setup and Protocol

Three pairs of commercially available MIMUs, each from a different manufacturer (from here on referred to as device model), were moved at three rotation rates. A detailed description of the experimental setup and the protocol used can be found in [9,10]. Briefly, three pair of MIMUs from Xsens—MTx (Xsens, Enschede, The Netherlands), APDM—Opal (APDM INC., Portland OR, USA.) and Shimmer—Shimmer3 (Shimmer Sensing, Dublin, Ireland) were aligned on a wooden board (Figure 1). The board was also equipped with eight reflective passive markers, three were used to define the Local Coordinate System (LCS) of SP while the remaining five were employed to enforce the ground-truth orientation by means of Singular Value Decomposition-based technique (SVD) [22]. The highly accurate alignment was ensured by using a T-square. The SP system consisted of 12 infrared cameras (Vicon T20, VICON, Yarnton, England). All the data were acquired using the proprietary software listed in Table 2. The synchronized dataset containing all the MIMU signals, the ground-truth orientation, and the videos of the performed movement (described in the following) can be found both on IEEE DataPort at http://dx.doi.org/10.21227/b23b-rx94 (accessed on 28 April 2021) [23], on GitHub (https://github.com/marcocaruso/mimu_optical_dataset_caruso_sassari, accessed on 4 May 2021), and also on MATLAB Exchange (https://it.mathworks.com/matlabcentral/fileexchange/91200-mimu_optical_dataset_caruso_sassari, accessed on 4 May 2021). Additional details about the synchronization process can be found in [10], Section 2.5.1. The noise level of the six MIMUs are reported in Table 3.

Before the acquisition of data, the MIMUs were turned on and warmed-up for 10 min to limit the influence of the temperature variation on the gyroscope measurements. Each recording started with a static period of 60 s (useful to compute the gyroscope biases), then the board was manually moved to span the three degrees of freedom of the space. The protocol was repeated at three rotation rates (rms 120 deg/s for a “slow” acquisition of 70 s, 260 deg/s for a “intermediate” acquisition of 45 s and 380 deg/s for a “fast” acquisition of 30 s). As an example, the Euler angles from the gold standard orientation were extracted for the intermediate trial to provide a visualization of the performed movements (Figure 2). In addition, it is possible to observe the entire experimental protocol by means of the videoclips uploaded with the dataset. The experiments were conducted under controlled conditions: the temperature was kept constant at 20 °C, while the ferromagnetic disturbances were limited, since the acquisitions took place one meter above the ground and within a 1 m^3^ of volume; the maximum variation of the magnetic norm was limited to 1 µT.

### 2.4. Data Processing

#### Orientation Estimation and Error Computation under Optimal and Suboptimal Conditions

The data processing described in this section has been performed entirely in MATLAB R2020a. The purpose of this section is to describe the grid-search procedure employed to obtain, for each SFA and for each of the nine experimental scenarios (i.e., 3 rotation rates × 3 device models), the absolute orientation error and the relative orientation difference for each combination of the two parameters being tuned. Bold quantities are intended to be matrices or vectors.

The following quantities are given:

$q_{SP}$ is the ground-truth orientation expressed in the quaternion form. It describes the orientation of the LCS of SP referred to its initial orientation and it was obtained by using the SVD technique [22]. From trigonometry considerations, as described in Section 2.5.1. of [10], the errors which affect the ground-truth orientation are lower than 0.5 deg;$p_{1vec}$ and $p_{2vec}$ are the two vectors which contain, for each SFA, the values of the two parameters ranging from 0 to $upper_{1}$ and from 0 to $upper_{2}$, respectively. In general, the two upper limits were chosen large enough to ensure that all the relevant search space was explored. The values of $upper_{1}$ and $upper_{2}$ can be observed in the figures of Appendix A. The lower limit for all the SFAs was set to zero but for a_th2_ of VAC which was set to the value of the first threshold for the accelerometer measurements (a lower value would be meaningless since for the constraint is a_th2_ ≥ a_th1_). The average number of combinations explored was not the same for all the SFAs since it was a trade-off between computational costs and the search space size (on average it amounts to 360 combinations).

Figure 3 shows the process described in this paragraph. The absolute orientation of two MIMUs (A and B) of the same manufacturer, for each combination of the values contained in $p_{1vec}$ and $p_{2vec}\;$was computed independently. Two quaternions $q_{A_{G}}$ and $q_{B_{G}}$ were obtained to describe the orientation of the LCS of each MIMU with respect to a GCS defined to have the vertical axis aligned to the gravity direction and one horizontal axis aligned with the Earth’s magnetic north projection on the horizontal plane. For this reason, to enable a proper comparison with $q_{SP}$, also $q_{A_{G}}$ and $q_{B_{G}}$ were referred to their initial orientation to obtain $q_{A}$ and $q_{B}$. This was possible thanks to the same alignment of the system LCSs on the board (Figure 1).

After that, the two absolute errors ($∆q_{abs\;A}$ and $∆q_{abs\;B}$) were obtained, separately for $q_{A}$ and $q_{B}$ by means of (1). The relative orientation difference ($∆q_{rel\;A,B}$) was also computed using (2). Symbols * and ⊗ denote the complex conjugate and the product in the quaternion algebra, respectively.
(1)$$∆q_{abs\;A}=q_{A}{}^{*}\;⊗q_{SP},\;∆q_{abs\;B}=q_{B}{}^{*}\;⊗q_{SP}.$$
(2)$$∆q_{rel\;A,B}=q_{A}{}^{*}\;⊗q_{B}.$$

To obtain a compact and interpretable representation of $∆q_{abs\;A}$, $∆q_{abs\;B}$ and $∆q_{rel\;A,B}$, the scalar part of each quaternion was converted into the corresponding rotation described using the axis-angle representation. Then, the rms of the dynamic parts of the recordings were considered to obtain a single value for each of the three quantities. The two absolute errors (in degrees) were then averaged to obtain $e_{i,j}$, which is the absolute average error for a given combination of the two parameter values. In the same way, the relative orientation difference corresponding to the same combination is called $\delta_{i,j}$. The matrices e
and δ were populated for each combination of the values contained in $p_{1vec}$ and $p_{2vec}$. All values were rounded to a 0.1 deg resolution.

### 2.5. Data Analysis

After having computed the e and δ matrices, the following steps assess the effectiveness of the RCM in providing suboptimal parameter values which lead to absolute errors close to the optimal. To this end, for a given SFA and for each experimental scenario, the first steps consisted in defining the optimal region and the corresponding optimal error starting from e. Then the suboptimal parameter values were extracted from δ. After that, the corresponding absolute error (i.e., suboptimal) was compared with the optimal one and the differences were analyzed. This process was repeated for each of the 90 cases (10 SFAs × 3 rotation rate magnitudes × 3 device models).

#### 2.5.1. Identification of the Optimal Region for Each Scenario and the Corresponding Optimal Absolute Error

For each SFA and for each experimental scenario, the two following optimal quantities were computed:

Optimal absolute orientation error: $e_{opt}=\min\left(e\right)$. In other words, $e_{opt}$ is the lowest error achievable when both parameter values are optimally tuned.The optimal region correspond to the range of the parameter values whose combinations provide errors within [$e_{opt}$, $e_{opt}$ + 0.5 deg], where 0.5 is the uncertainty related to the ground-truth errors: $\left\{p_{opt_{1}},p_{opt_{2}}\}=\{\left(p_{1vec},p_{2vec}\right)\;|\;e\;\leq e_{opt}+0.5\;deg\right\}$.

#### 2.5.2. Identification of the Suboptimal Parameter Values for Each Scenario and the Corresponding Suboptimal Absolute Error

For each SFA and for each experimental scenario, the following suboptimal quantities were computed:

Minimum relative orientation difference: $\delta_{sub}=\min\left(\delta \right)$.The suboptimal region is defined by the values of $p_{1\mathrm{vec}}$ and $p_{2vec}$ corresponding to $\delta_{sub}$: $\left\{p_{sub_{1}},p_{sub_{2}}\right\}=\{\left(p_{1vec},p_{2vec}\right)\;|\;\delta =\delta_{sub}\}.$ When the region, $\left\{p_{sub_{1}},p_{sub_{2}}\right\}$ was formed by two or more separated sub-regions, only the largest was considered.The suboptimal parameter values ($p_{1c}$ and $p_{2c}$) are the values of $p_{1vec}$ and $p_{2vec}$ corresponding to the centroid of the suboptimal region: $\left\{p_{1c},p_{2c}\right\}=\mathrm{centroid}\;\left(p_{sub_{1}},p_{sub_{2}}\right)$.The suboptimal absolute orientation error is the absolute orientation error corresponding to $p_{1c}$ and $p_{2c}$: $e_{sub}=$ $e\left(p_{1c},p_{2c}\right)$.

#### 2.5.3. RCM Validation Metric

For each of the 90 combinations, the residual between the optimal and suboptimal error was computed as follows:(3)$$∆e=e_{sub}-e_{opt}.$$

The distributions of the 90 residuals were then analyzed using a boxplot representation. As previously stated, when ∆e was not larger than 0.5 deg, the RCM was considered completely effective.

## 3. Results

### 3.1. Optimal and Suboptimal Errors

The minimum orientation error ($e_{opt}$), the suboptimal orientation error ($e_{sub}$), and their residual (∆e) are listed in Table 4 for each SFA and experimental scenario. The residual value for ∆e exceeding 0.5 deg is reported in bold. The distribution of the 90 values of ∆e is represented by means of a boxplot in Figure 4.

### 3.2. Optimal and Suboptimal Regions

The optimal and suboptimal regions corresponding to the nine experimental scenarios are reported in two figures for each SFA in Appendix A. Moreover, for each suboptimal region, the centroid is also indicated with a circular marker. For the SFA with only one parameter tuned the regions were represented as a 1D interval (instead of a bidimensional region).

Figure 5 provides an example of both optimal and suboptimal regions for the LIG filter. The regions are highlighted with a different color for each experimental scenario.

## 4. Discussion

One of the most important steps when using a sensor fusion algorithm is the choice of the value(s) of filter parameter(s) [2,9,24,25] since they heavily affect the orientation accuracy. As discussed in [10], a possible choice is to perform a fine-tuning of these values specifically for a given experimental scenario using the ground-truth orientation. The errors obtained this way are representative of the best performance achievable with a given SFA. Furthermore, it has been observed that there is not a common intersection among the optimal regions when varying the experimental scenario (see Appendix A “Optimal regions (intervals)”). This stresses the importance of tuning the parameter values of each SFA according to the specific scenario under analysis (device model and rotation rate).

In this work, suitable parameter values were estimated using the RCM method, which does not rely on any ground-truth orientation to reflect a more common situation during the everyday use of the MIMUs. Table 4 shows that the residuals between the optimal and suboptimal errors were lower or equal to 0.5 deg in 60 cases out of 90, between 0.5 and 1.0 deg in 10 cases, and higher than 1.0 deg in the remaining 20 cases. Thus, the RCM allowed to estimate the orientation with errors equivalent to the optimal approach in 67% of the cases. In the remaining 33%, the maximum residual amounted to 3.7 deg. Overall, the median residual was equal to 0.2 deg and the computed mean to 0.6 deg. These results corroborate the findings obtained by Caruso et al., 2020 [9] for a single SFA and suggest the possibility to properly tune a generic SFA on different scenarios without using any orientation reference.

Among the 30 cases in which ∆e was higher than 0.5 deg, an increase of the rotation rate magnitudes led to inferior performance of RCM. In fact, 14/30 cases were at fast rotation rate vs 12/30 at intermediate and only 4/30 at slow. This is in line with the unfavorable effect of the rotation rate in the orientation estimation accuracy, as widely recognized in the literature [2,9,10,26,27]. Moreover, also the specific device model had an influence on the accuracy of the RCM. In the 30 cases in which ∆e was higher than 0.5 deg, 14 were associated with Shimmer, 11 with APDM while 5 with Xsens. Finally, the LIG was the SFA for which the RCM performed the best. In fact, only 1 out of 9 residuals was higher than 0.5 deg while 5 out of 9 residuals were higher than 0.5 deg for MAH and VAK. The four residuals marked as outliers in Figure 4 were equal to 2.4 deg, 3 deg, 3.3 deg and 3.7 deg and were obtained with Shimmer at intermediate and fast rotation rates.

The figures shown in Appendix A “Suboptimal regions (intervals)” suggest that the suboptimal regions do not consist of a single point. This is because multiple combinations of the two parameter values provide the same minimum of the relative orientation difference. However, it should be highlighted that the absolute errors corresponding to these combinations may be different.

Some limitations must be considered when using the RCM. Since the method relies on the differences between the errors affecting the two different accelerometers and magnetometers, when their mutual distance approaches zero also the differences tend to be less evident. In this case, the relative orientation difference may be very small, but it does not guarantee low absolute errors, especially if the orientations of the two MIMUs are estimated giving a high weight to the accelerometer and magnetometer readings. The authors suggest placing the two MIMUs with a mutual distance of at least a few centimeters, compatibly with the size of the rigid body support.

Some applications can benefit from this approach. In fact, as described in [28] and detailed in [9] a miniaturized plastic case may be designed for each specific application to host two MIMUs and to rigidly attach them to the body segment of interest. If only one MIMU is necessary for the data collection, the miniaturized case may host both the MIMU to be employed and an additional MIMU and a preliminary movement acquisition which mimics the gesture under the same experimental scenarios (similar rotation rate magnitude and device model) may be performed to estimate the suboptimal parameter(s) of the selected SFA.

## 5. Conclusions

The RCM originally proposed in [9] was tested on various SFAs to provide the suboptimal parameter values by exploiting the hypothesis that the orientation of two MIMUs fixed to the same rigid body and aligned with each other remains constant. To the best of authors’ knowledge, the RCM is the only published method that does not rely on the knowledge of any ground-truth orientation to estimate the parameter values. The results are encouraging, since the maximum residual between the optimal error and the error obtained using the estimated suboptimal parameter values amounted to 3.7 deg and to 0.6 deg on average. Moreover, the results confirmed the absence of a unique set of parameter values suitable for all the experimental scenarios and also that a proper tuning of the parameter values is necessary for all SFAs to obtain acceptable estimates of the absolute orientation. The findings of the present study may be used to develop additional methods to estimate the parameter values with only one MIMU.

## Figures and Tables

**Figure 1 sensors-21-06307-f001:**
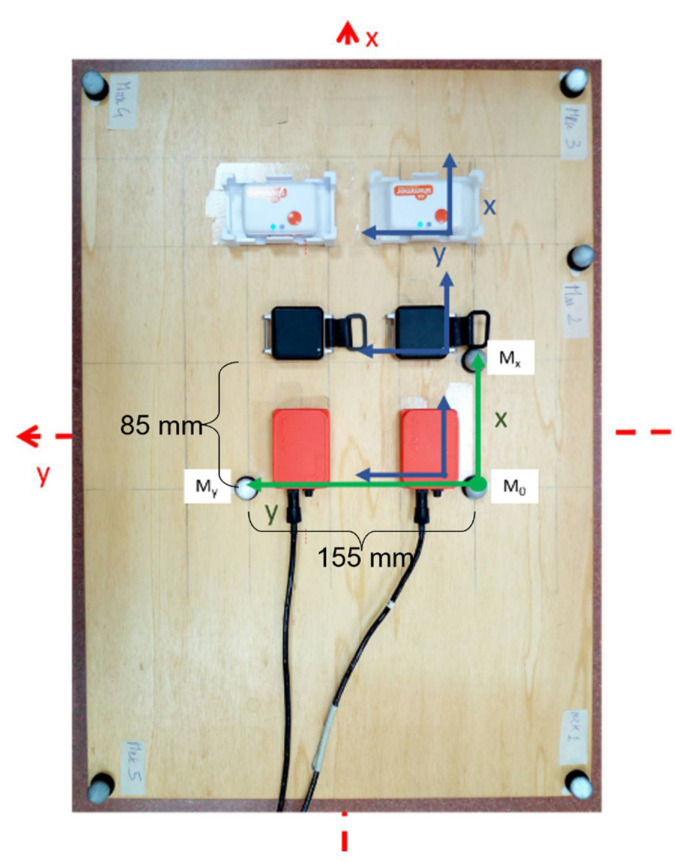
The experimental setup employed. Three pairs of MIMUs were aligned (from the bottom: Xsens—MTx, APDM—Opal and Shimmer—Shimmer3). The LCSs of the MIMUs are represented in blue. The three central markers define the LCS of the SP (in green). The MIMUs and SP systems were arranged so that their axes were aligned with the axes of the board (dashed red arrows). Figure adapted from [9].

**Figure 2 sensors-21-06307-f002:**
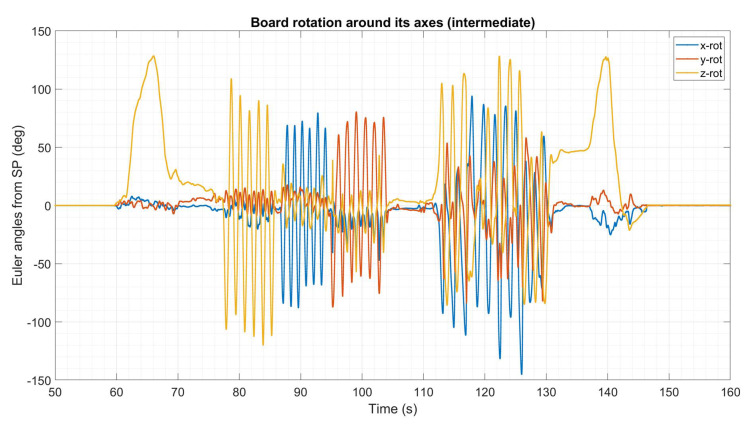
Exemplificative description of the movements of the board in terms of Euler angles for the intermediate trial. As evident, from the graph, the first three rotations were performed around one axis at a time, while the last part of the movement is a combination of the movement around the three axes.

**Figure 3 sensors-21-06307-f003:**
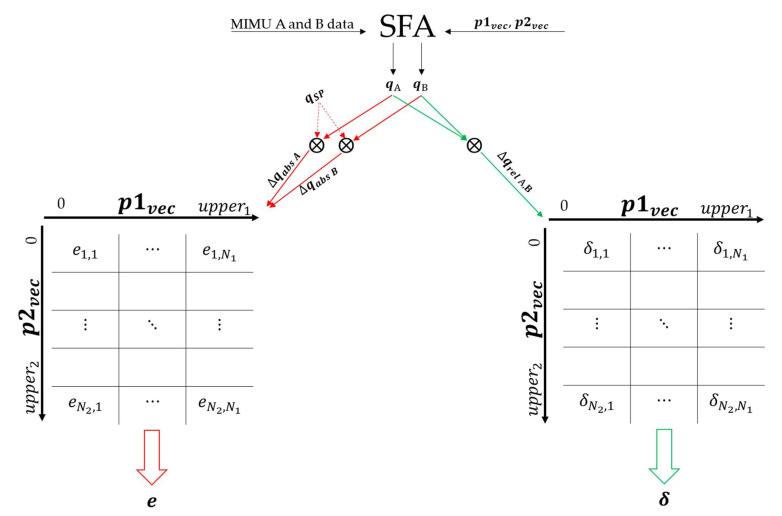
The grid-search approach followed to compute the absolute orientation error and the relative orientation difference for a given combination of the two parameter values. This process has been applied to each SFA for each of the nine experimental scenarios. Red and green arrows are related to the computation of the absolute error and relative difference, respectively.

**Figure 4 sensors-21-06307-f004:**
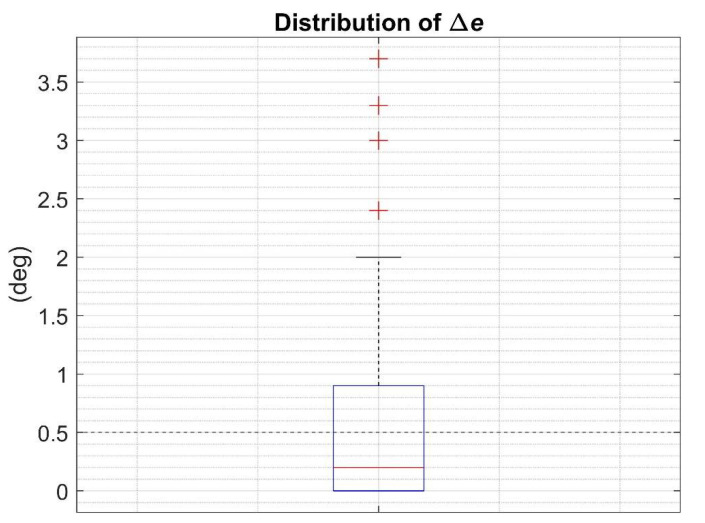
Boxplot of the distribution of the 90 residuals (∆*e*). Outliers are also reported(red cross). The limit of 0.5 deg chosen to consider the suboptimal errors equivalent to the optimal error is also highlighted.

**Figure 5 sensors-21-06307-f005:**
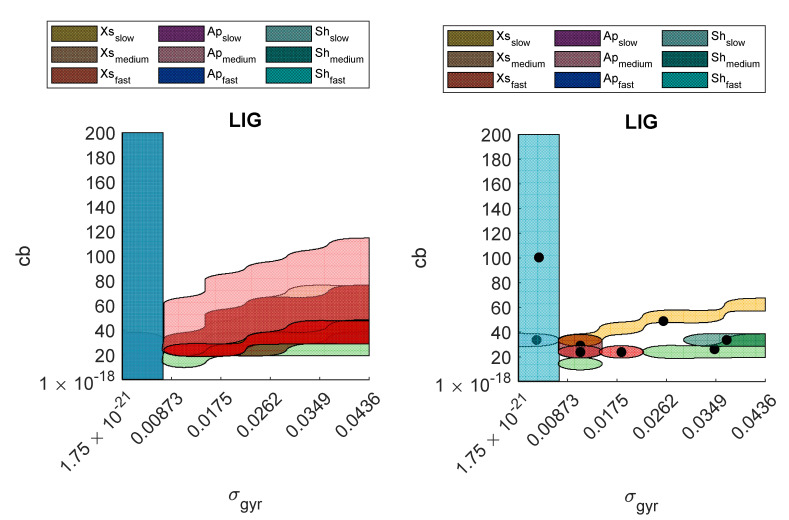
On the left: the optimal regions (one for each experimental scenario) obtained for LIG. On the right: the suboptimal regions (one for each experimental scenario) and their centroids obtained for LIG.

**Table 1 sensors-21-06307-t001:** This table reports the details of the two parameters selected for optimal and suboptimal tuning along with their default values. Adapted from [10].

CF	# Params	$p_{1}$	Default		$p_{2}$	Default	
MAH	2	k_p_—inverse gyroscope weight	1	rad/s	k_i_—weight for online bias estimation	0.3	rad/s
MAD	1	β—inverse gyroscope weight	0.1	rad/s	/	/	
VAC	9	g_mag_—magnetometer weight	0.01	a.u.	a_th2_—threshold for accelerometer vector selection	0.2	a.u.
SEL	4	τ_acc_—accelerometer time constant	1	s	τ_mag_—magnetometer time constant	3	s
MCF	2	g_mag_—magnetometer weight	0.01	a.u.	/	/	

**KF**	**# Params**	** $p_{1}$ **	**Default**		** $p_{2}$ **	**Default**	
SAB	6	σ_gyr_—inverse gyroscope weight	0.007	rad/s	a_th_—threshold for accelerometer vector selection	40	mg
LIG	6	σ_gyr_—inverse gyroscope weight	1	rad/s	c_b_—Gauss-Markov parameter of the prediction model to set the variance of external acceleration and ferromagnetic disturbances	1	a.u.
VAK	3	σ_gyr_—inverse gyroscope weight	0.004	rad/s	σ_acc_—inverse accelerometer weight	0.014	m/s^2^
GUO	3	σ_gyr_—inverse gyroscope weight	0.001	rad/s	/	/	
MKF	8	σ^2^_gyr_—inverse gyroscope weight	9.14 × 10^−5^	(rad/s)^2^	/	/	

**Table 2 sensors-21-06307-t002:** Details of data acquisition for each system used.

System	Software	Sampling Frequency
Xsens—MTx	MT Manager Version 1.7	100 Hz
APDM—Opal	Motion Studio Version 1.0.0.201712300	128 Hz (resampled at 100 Hz)
Shimmer—Shimmer3	Consensys v.1.5.0	100 Hz
Vicon—T20	Nexus v2.7	100 Hz

**Table 3 sensors-21-06307-t003:** The STD of the three sensors computed for each of the six MIMUs during one minute of static acquisition. Taken from [10].

STD	Accelerometer (mg)			Gyroscope (deg/s)			Magnetometer (µT)		
	x	y	z	x	y	z	x	y	z
Xsens-MTx #1	0.86	0.80	0.85	0.38	0.39	0.37	0.06	0.04	0.04
Xsens-MTX #2	0.82	0.86	0.80	0.44	0.40	0.40	0.05	0.06	0.06
APDM-OPAL #1	0.38	0.33	0.38	0.16	0.23	0.11	0.26	0.23	0.20
APDM-OPAL #2	0.34	0.32	0.35	0.16	0.27	0.19	0.26	0.25	0.20
Shimmer-Shimmer 3 #1	1.06	0.97	1.26	0.09	0.08	0.09	0.84	0.84	0.69
Shimmer-Shimmer 3 #2	1.12	1.09	1.29	0.06	0.06	0.06	0.97	0.97	0.58

#1 and #2 denote the two units of the same commercial product.

**Table 4 sensors-21-06307-t004:** The optimal and suboptimal errors are reported together with their residual for each SFA and for each experimental scenario. The residual values higher than 0.5 deg are highlighted in bold.

		CF	$e_{opt}$	$e_{sub}$	∆e	KF	$e_{opt}$	$e_{sub}$	∆e
Xsens	Slow	MAH	2.5	2.5	0	SAB	2.2	2.2	0
	Intermediate		2.4	3.8	1.4		2.1	2.1	0
	Fast		4.0	4.2	0.2		2.4	2.4	0
APDM	Slow		3.8	5.6	1.8		5.0	5.1	0.1
	Intermediate		4.8	4.9	0.1		5.7	5.8	0.1
	Fast		8.2	9.2	1		8.3	10.0	1.7
Shimmer	Slow		3.4	3.7	0.3		4.5	4.5	0
	Intermediate		4.6	5.3	0.7		4.9	4.9	0
	Fast		7.6	10.6	3		8.5	9.6	1.1
Xsens	Slow	MAD	2.7	2.7	0	LIG	1.9	2.4	0.5
	Intermediate		2.5	4.0	1.5		2.0	3.8	1.8
	Fast		4.0	4.0	0		2.9	3.4	0.5
APDM	Slow		3.8	3.8	0		3.6	3.9	0.3
	Intermediate		4.6	4.8	0.2		4.9	5.1	0.2
	Fast		8.1	8.2	0.1		4.6	4.9	0.3
Shimmer	Slow		3.9	4.1	0.2		4.4	4.6	0.2
	Intermediate		4.9	5.1	0.2		4.0	4.2	0.2
	Fast		8.8	10.8	2		6.3	6.5	0.2
Xsens	Slow	VAC	4.0	4.0	0	VAK	1.2	1.5	0.3
	Intermediate		5.0	5.1	0.1		1.6	1.7	0.1
	Fast		7.2	7.2	0		2.5	2.5	0
APDM	Slow		3.5	4.4	0.9		3.6	4.1	0.5
	Intermediate		6.1	6.4	0.3		6.0	6.9	0.9
	Fast		8.3	11.3	3		9.2	10.4	1.2
Shimmer	Slow		3.8	3.8	0		4.0	4.6	0.6
	Intermediate		10.2	10.8	0.6		4.4	5.7	1.3
	Fast		11.5	15.2	3.7		8.2	10.6	2.4
Xsens	Slow	SEL	3.1	3.5	0.4	GUO	2.3	2.3	0
	Intermediate		2.5	3.9	1.4		2.3	2.3	0
	Fast		5.1	5.1	0		5.7	5.7	0
APDM	Slow		3.7	3.8	0.1		4.2	4.5	0.3
	Intermediate		7.1	7.1	0		5.1	5.7	0.6
	Fast		8.0	10.0	2		9.4	9.4	0
Shimmer	Slow		3.4	3.5	0.1		4.0	4.2	0.2
	Intermediate		5.0	6.3	1.3		5.1	5.1	0
	Fast		9.4	10.8	1.4		13.7	14.4	0.7
Xsens	Slow	MCF	3.3	3.4	0.1	MKF	4.2	4.3	0.1
	Intermediate		6.1	6.1	0		4.8	4.8	0
	Fast		6.6	7.5	0.9		6.7	6.9	0.2
APDM	Slow		3.8	4.8	1		3.6	3.8	0.2
	Intermediate		12.3	12.5	0.2		5.3	5.3	0
	Fast		7.9	9.6	1.7		7.2	7.2	0
Shimmer	Slow		5.0	5.2	0.2		3.9	4.2	0.3
	Intermediate		10.0	13.3	3.3		8.4	9.8	1.4
	Fast		8.6	8.8	0.2		9.9	10.0	0.1

## Data Availability

The sensor fusion algorithm together with the optimization codes are available as MATLAB functions and scripts at https://github.com/marcocaruso/sensor_fusion_algorithm_codes, accessed on 4 May 2021 and also on MATLAB Exchange at https://it.mathworks.com/matlabcentral/fileexchange/90351-orientation_estimation_sensor_fusion_algorithm_codes, accessed on 4 May 2021. The complete dataset is available on both IEEE DataPort at http://dx.doi.org/10.21227/b23b-rx94 (accessed on 28 April 2021) [23], on GitHub (https://github.com/marcocaruso/mimu_optical_dataset_caruso_sassari, accessed on 4 May 2021) and also on MATLAB Exchange (https://it.mathworks.com/matlabcentral/fileexchange/91200-mimu_optical_dataset_caruso_sassari, accessed on 4 May 2021).

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
