# Peer review of "Extension of the Rigid-Constraint Method for the Heuristic Suboptimal Parameter Tuning to Ten Sensor Fusion Algorithms Using Inertial and Magnetic Sensing"

_sensors, 2021, doi:10.3390/s21186307_

Round 1

Reviewer 1 Report

General

This paper evaluates an earlier presented method for tuning parameters of 3D orientation estimation algorithms using MIMU sensors without the need to use a lab-bound reference measurement. In the current paper, this method is evaluated on 10 different sensor fusion algorithms presented in literature. This novel study is very relevant, since tuning orientation estimation algorithms without the need for lab-bound reference measurement systems is important, when applying these algorithms outside the lab in daily-life.

Main comments/questions:

  1. I wonder whether the basic assumption on which the RCM method is based is valid in general. The arguments that you provide are not completely clear to me (Please explain more clearly, provide clear arguments and indicate limitations) :

-  ln. 50-54: ‘The assumption on which the method relies, is that two MIMUs aligned on a rigid body must have a null orientation difference during the movement, as the error and disturbance characteristics affecting the sensors embedded in the two MIMUs are, in general, independent, as discussed in more detail in section 2.1.1.’..

- ln. 90-95: ‘The RCM assumption is that the combination of parameter values that provide the minimum relative orientation difference also provide an acceptable absolute orientation error…. The hypothesis under analysis holds when the source of errors affecting gyroscope, accelerometer, and magnetometer are different between the MIMUs….’

- ln 358-359: ‘the authors suggest placing the two MIMUs with a mutual distance of at least a few centimeters, compatible with the size of the target.’(which target?)

Specific comments/questions:

  1. 2, ln. 47-50: you refer to your previous work ([9]?), but only references to other papers [6],[12] are specified. Specify [9] at here?
  2. 1, pg. 5: the 2 sensors of each brand were not placed on the same position on the board: were the distances of the sensors in the y-direction (as indicated in fig.1) such that the imposed rotational movements (rotational accelerations, angular velocities) imposed in the experimental data induced a relevant difference in accelerometer signals? The imposed angular velocities are reported (ln. 183-185), but the distance between the sensors are not reported. Please specify this distance. In ln. 358-359 you suggest placing the MIMUs with a mutual distance of at least a few centimeters. Please specify your argument for this in the Discussion.
  3. 6, ln 182: what is meant with ‘the board was manually moved to span the three degrees of freedom of the space.’? Were the 3 indicated angular velocities implemented around a variety of representative axis directions or only around the x, y, z axes indicated in fig. 1?
  4. Table 1: why is no second varied parameter (p2) indicated for the 2nd, 5th, 9th and 10th algorithms listed? Does this mean that for these algorithms only 1 parameter was estimated using suboptimal tuning? Does this result in a fair comparison between all 10 algorithms?
  5. 7: ln. 232-239. This section is not completely clear to me. I agree that the orientation errors can be represented by the scalar part of the delta quaternions specified in ln 232-233. What do you mean by ‘The two angular absolute errors’ in ln. 236?
  6. 8, 2.5.2: ln 273-276: I miss specification of the allowed deviation of delta_sub to determine the suboptimal region. In comparison, you specified the optimal region in section 2.5.1 being the region where the error deviated max 0.5 deg from the minimum error.
  7. 8, ln. 291: why specify ‘minimum orientation error e_opt, and not MINIMAL suboptimal orientation error e_sub? If not considering the minimum suboptimal orientation error, the suboptimal error would be a range and not one value, as specified in table 4.
  8. Do you want to provide the relevant, but extensive, Appendix A as part of the paper or as supplementary material, which is reachable using a link specified in the paper? You may want to check the policy of the Sensors journal.
  9. 4 (and Appendix A): not all 9 colours can be distinguished in the graphs (see fig. 4, e.g. purple yellow scenario in left figure). Probably, this is due to the overlap of regions for different scenarios. How did you represent this? It seems that you use mixed colours, which may not be very clear. Please optimize and inform the reader.
  10. 11, ln. 325: the term RCM (rigid constraint method) has been used throughout the paper. I understand that it refers to the suboptimal parameter estimation method without using a reference. Please specify why you use the term rigid constraint method for this. This has not become clear to me from reading this paper. The first reference to RCM (ln. 48) refers to your previous paper [9]
  11. You may want to discuss that the effect of magnetic disturbances (which is often of practical relevance when including the use of magnetometers) were not considered when tuning sensor parameters in the optimal / suboptimal regimes.

Reviewer 2 Report

GENERAL COMMENTS:

This paper proposed a Rigid-Constraint Method (RCM) to estimate suboptimal parameter values of MIMU based sensor fusion algorithms without relying on any orientation reference. It aimed at solving the troublesome but necessary parameter tuning issue in sensor fusion algorithms, which directly contribute new knowledge to the scientific community. The authors analyzed the most popular Complementary filters and Kalman filters, and also make them open source. The methods and conclusions are clear. The introduction can be improved, and some specific issues need the authors' attention.

SPECIFIC COMMENTS:

1.    It seems uncommon to include so many keywords after the abstract.
2.    The introduction can be improved. It contains several very long sentences (Line 33 to line 38). Line 57 to 59 stated the RCM is a good method but missing some statements that lead to this study. It is not a good style to mix many references together, e.g. [1-7], [8-11].
3.    It would be beneficial to add a figure of Euler angles of a representative trial, to show the details of how you performed the experiments.
4.    Line 88: the most important method (RCM) was proposed in the previous publication [9].  Suggest to clearly state the contributions of this paper in comparison with previous publications ([8],[9],[10])? 
5.    Line 151: “two MIMUs from Xsens” is not clear. “Three pairs of MIMUs” in line 170 is better.
6.    Line 164: How to synchronize the three IMU systems and Vicon system? 
7.    Figure 1: Why need three pairs of MIMU from different manufacturers? Which sensor raw signal is the best?
8.    Figure 1: Did you perform Magnetic Field Mapper for XSENS sensor and other sensors. As the nearby sensor contain hard iron disturbance, which should be removed before testing.
9.    Table 4: some data “4,0”, should the comma be a point ?
10.    Line 371: This sentence is confusing, the RCM originally proposed in [9] not this paper.
11.    Line 378: Is it a future work? For the current RCM method, how to use it in real-scenario? Do it required two IMU tie together. If it is, it still needs a extra device and a special setup.

Reviewer 3 Report

This paper is of great importance to enhance the attitude precision of MIMU estimated by SFA.

This paper proposed a Rigid-Constraint Method (RCM) of estimating suboptimal parameter values without relying on any orientation reference. Its effectiveness and applicability of the RCM were evaluated and tested on some experiments.

There are some problems in the paper, for example:

(1)In line 15,”can be ”->”is”?

(2)Line 21, “any orientation reference” ->” any orientation references”?

(3) Line 173,“axes f” ?

(4) Line 287, “lower or equal than”->” lower than or equal to”? or “not larger than”?

(5) Figures in this paper are not expounded clearly.
